# Rapid Detection of Changes in Image Textures of Carrots Caused by Freeze-Drying using Image Processing Techniques and Machine Learning Algorithms

Ewa Ropelewska [1,*], Kadir Sabanci [2], Muhammet Fatih Aslan [2] and Necati Çetin [3]

1 Fruit and Vegetable Storage and Processing Department, The National Institute of Horticultural Research, Konstytucji 3 Maja 1/3, 96-100 Skierniewice, Poland
2 Department of Electrical and Electronics Engineering, Karamanoglu Mehmetbey University, Karaman 70100, Turkey; kadirsabanci@kmu.edu.tr (K.S.); mfatihaslan@kmu.edu.tr (M.F.A.)
3 Department of Agricultural Machinery and Technologies Engineering, Faculty of Agriculture, Ankara University, Ankara 06110, Turkey
* Correspondence: ewa.ropelewska@inhort.pl

**Abstract:** The objective of this study was to evaluate the differences in texture parameters between freeze-dried and fresh carrot slices using image processing and artificial intelligence. Images of fresh and freeze-dried carrot slices were acquired using a digital camera. Texture parameters were extracted from slice images converted to individual color channels *L*, *a*, *b*, *R*, *G*, *B*, *X*, *Y*, and *Z*. A total of 1629 texture parameters, 181 for each of these color channels, were obtained. Models for the classification of freeze-dried and fresh carrot slices were created using various machine learning algorithms, based on attributes selected from a combined set of textures extracted from images in all color channels (*L*, *a*, *b*, *R*, *G*, *B*, *X*, *Y*, and *Z*). Using three different feature selection methods (Genetic Search, Ranker, and Best First), the 20 most effective texture parameters were determined for each method. The models with the highest classification accuracy obtained by applying various machine learning algorithms from Trees, Rules, Meta, Lazy, and Functions groups were determined. The classification successes obtained with the parameters selected from all three different feature selection algorithms were compared. Random Forest, Multi-class Classifier, Logistic and SMO machine learning algorithms achieved 100% accuracy in the classification performed with texture features obtained by each feature selection algorithm.

**Keywords:** carrot slices; image analysis; texture parameters; classification; artificial intelligence





## 1. Introduction

A carrot, also known as *Daucus carota* L., is an important vegetable that is consumed at a high rate worldwide. It is among the ten most important vegetable types, in terms of both production and marketing [1,2]. According to the 2022 data published by Atlas Big [3], more than 44 million tons of carrots and turnips are produced annually in the world. The country that grows the most carrots in the world is China, with 21.5 million tons. The countries that produce the most carrots after China are Uzbekistan, the United States, Russia, and Ukraine, respectively. The carrot belongs to the Apiaceae family, which also includes vegetables such as fennel and celery. Like most members of this family, carrots are aromatic. They are widely used as a spice, or for medicinal purposes [2].

The carrot is a root vegetable, rich in dietary fibers and bioactive compounds, such as carotenoids. Carrots, and carrot-derived products, also have antioxidant content that reduce the effects of cancer disease. Carrot pulp contains 50% β-carotene, which is a very important micronutrient for human health. Different carotenoids, including β-carotene, are of great interest today for their possible protective effects against some types of cancer [4,5]. Carrots are also rich in vitamins A, B, and C, and minerals. They are a source of many

minerals and carbohydrates, such as Fe, Mg, Ca, etc. [6]. In addition to being consumed as a vegetable, carrots are used in various fields, such as fruit juice production, baby food production, pickle, canned food, cake-making, etc. [7]. The orange color of the carrot, which has many different types, such as black, purple, white, orange, etc., is consumed more worldwide. Orange-colored carrots are very rich in carotene and vitamin A. In addition to their different colors, there are many different types of carrot shapes, such as long, thin, thick, and round. Nutritional properties do not change according to the shape of the carrot. However, the shape of the carrot is of great importance in terms of marketing. Customers generally do not prefer the irregularly shaped carrots in the market. Keeping the carrot in an open environment for a while reduces the quality of the carrot. For this reason, if carrots with an irregular shape remain in the markets for a while, it leads to product loss [8].

High moisture content makes agricultural products susceptible to spoilage [9]. Similarly, carrots contain a high percentage of moisture. Drying is the most effective method for prolonging the life of the carrot for off-season or long-term use. In this way, it is aimed at minimizing the biochemical, chemical, and microbiological deterioration of the product [10]. In the food industry, many drying methods are used to remove moisture from fresh products in order to extend their shelf life [11]. For this purpose, most agricultural products use air convective methods. Since it is a product exposed to open environments during open-air sun-drying, it can be easily polluted by dust, insects, rodents, and birds [12]. Although air convective drying is common and easy to apply, low product quality (loss of color and aroma) and high energy consumption limit the use of this technique [13]. In addition, in the convective method, shrinkage and loss of nutrients and flavor may occur [14]. Microwave drying may cause over-browning of fruits and loss of taste, due to exposure to excessive heat [15]. For this reason, freeze-drying, which offers the best final product quality among drying methods, comes to the fore. Freeze-drying (FD) is considered to be the drying technique that yields the highest quality dried products, especially for sensitive surface products that are adversely affected by conventional drying processes [16]. In contrast to convective drying, where undesirable microstructural and sensory changes are inevitable in freeze-drying, they regain most of their original texture when rehydrated, with minimal shrinkage [17]. FD occurs in two stages. The first freezing step has a major impact on the overall efficiency of the freeze-drying process, because it determines the structure of the ice crystals, which affects the heat and mass transfer rates and subsequent sublimation (drying) steps. In the second stage, sublimation, the ice crystals should be large, to reduce the drying time in the primary drying period; however, in the second drying stage, it is desirable to have the smallest possible crystals, in order to increase the specific surface area of the pores. All these trends lead to the existence of optimum pore size [18,19]. In this method, due to the absence of liquid water and the low temperatures required for freeze-drying, most spoilage and microbiological reactions are delayed, resulting in a high-quality final product [20]. However, freeze-dried carrots have been reported to have a higher carotenoid content, a robust microstructure, and a brighter color, compared to hot-air-dried carrots [2].

A wide range of applications have been conducted on carrots using freeze-drying processes or methods [2,14,20–23]. Compared to alternative drying processes, freeze-drying is very advantageous in terms of preserving color, taste, aroma, and nutrients [21]. However, in the study by Fan, et al. [22], it was emphasized that there were differences in color and texture in unprocessed and processed (freeze-drying) carrot slices. Similarly, Rajkumar, et al. [23] analyzed the change in the physical properties of carrots caused by hot air and freeze-drying processes. Although not as much as hot-air-drying, it was stated that the freeze-drying process caused color and textural differences in carrots. Such color and texture changes in dried fruits and vegetables significantly affect the quality and market value of the product. Manually detecting and analyzing these changes is difficult and leads to biased results. The use of computer-aided systems to detect color and textural changes provides more objective inferences about the freeze-drying process. Textural features are formed by the spatial distribution of pixels. Changes in both color and texture after freeze-drying can be successfully analyzed by image processing techniques that provide

pixel-based processing [9]. Artificial intelligence-based methods are popular today to automatically perform these pixel-based analyses. In this context, different machine learning and deep learning-based studies [24–28] are frequently used to distinguish different agricultural products. Artificial intelligence contributes to the rapid digital transformation and growing power of the agriculture industry. Computer vision is a field of artificial intelligence representing the human visual system using models. Intelligent systems based on computer vision are well utilized for agricultural operations. The combination of high-quality image acquisition and computer vision techniques is a cutting-edge approach that enables efficient technology-driven solutions in agriculture [29]. Agriculture ecosystems need to be constantly monitored. The processing of resulting data, in the form of images with image algorithms by machine learning, has great potential for future agricultural needs. Particularly, deep learning has attracted extensive attention in computer vision and image processing applications, and is characterized by an increasing potential in the agriculture sector [30]. Remarkable results in different sectors of agriculture can be related to machine learning techniques applied to texture, shape, color, and spectral analysis of object images. Current research dynamics in machine learning technologies for agricultural machine vision systems indicate their expanding application in precision agricultural systems for feature data analysis [31]. The emergence of machine vision technology resulted in overcoming many limitations of the classification and detection of food products using other methods [32]. The application of computer vision and image processing allows for the elimination of subjective and time-consuming industrial food quality control. Therefore, the application of these technologies to assess and improve the quality of foods has increased significantly [33]. In our own previous studies, in the case of the processing of carrots, machine learning turns out to be useful also for the classification of lacto-fermented and fresh carrot slices, based on texture parameters [25].

The objective of this study was to develop an innovative approach to classifying freeze-dried and fresh carrot slices using selected image texture parameters and machine learning algorithms. This is the first study that analyzes the change in freeze-drying processed carrot products with artificial intelligence-based methods, in terms of textures selected from a large set of approximately 1600 features from images in color channels *L*, *a*, *b*, *R*, *G*, *B*, *X*, *Y*, and *Z*, which is the great novelty in evaluating differences between fresh and dried carrot. Different feature selection methods were used to choose textures with the highest power for distinguishing the carrot samples. Furthermore, various machine learning algorithms from Trees, Rules, Meta, Lazy, and Functions groups were applied. Therefore, the main contributions of this study are:

- detection of changes in carrots caused by freeze-drying in a non-destructive and objective manner;
- distinguishing fresh and dried carrot samples, based on selected image textures;
- assessment of the usefulness of textures selected using various selection methods for the classification of fresh and dried carrot samples;
- comparison of the effectiveness of machine learning models built using different algorithms.

## 2. Materials and Methods

### 2.1. Materials

The carrot roots were sampled from a backyard garden located in north-eastern Poland. After obtainment, roots were washed and drained of excess water. Eighty fully developed, undamaged roots were selected for experiments. The roots were peeled and sliced using a sharp knife. In the case of each carrot root, five slices of 5 mm thickness were obtained. Thus, the research material comprised of four hundred carrot slices in total (200 slices subjected to direct imaging in their fresh form, and 200 slices intended for freeze-drying and then imaging as freeze-dried ones). The sample whole fresh carrot root and extracted fresh carrot slices are presented in Figure 1.

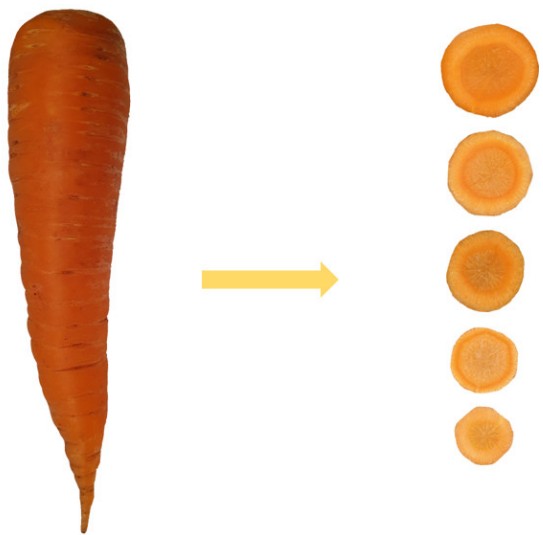

**Figure 1.** The exemplary whole fresh carrot root and fresh carrot slices.

The carrot slices intended for freeze-drying were frozen at a temperature of −28 °C ± 1 °C in a freezer (Whirlpool) for 2 days, before drying. The freeze-drying was performed for 48 h using a freeze dryer (LABCONCO, Kansas City, MO, USA). Samples were placed on shelves at a temperature of 20 °C. At the beginning, the process was carried out without heating, the pressure was equal to 5 mBar, and the temperature of the condenser was −55 °C. After 2 h, a pressure of 0.140 mBar, and temperature of the condenser of −53 °C were observed. At the final stage of the process, the temperature of the shelves reached 30 °C, and the final pressure was equal to 0.100 mBar. Similar to the fresh samples, the obtained freeze-dried slices were also imaged. The block diagram of the proposed approach to the classification of freeze-dried and fresh carrot slices, based on selected image features, is presented in Figure 2.

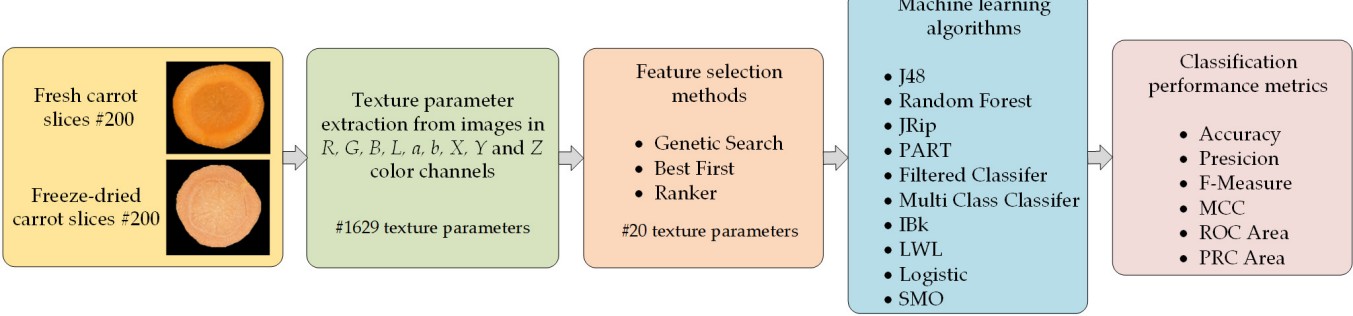

**Figure 2.** Block diagram of the classification of freeze-dried and fresh carrot slices, using image processing and machine learning.

## 2.2. Image Processing

Both fresh and freeze-dried carrot slices were subjected to color imaging using a digital camera and LED (Light-Emitting Diode) illumination. Fresh samples were drained of surface fluid using a paper towel. Samples were imaged on a black background. Each image contained twenty carrot slices (Figure 3). In total, 10 images with 20 fresh carrot slices each, and 10 images with 20 freeze-dried carrot slices each, were obtained. After acquiring, the images were uploaded to the computer with Microsoft's Windows operating system, using the USB cable. The images were saved in the BMP file format, which allowed for further image processing using MaZda software (Łódź University of Technology, Institute of Electronics, Łódź, Poland) [34–36]. At first, the carrot slice images were converted to

color channels *R*, *G*, *B*, *L*, *a*, *b*, *X*, *Y*, and *Z*. The black background facilitated the image segmentation and the separation of lighter carrot slices as one ROI (region of interest) for each slice. The image segmentation and carrot slice separation from the background were carried out based on the pixel brightness intensity. In the case of each ROI, 181 texture parameters based on the run-length matrix, histogram, co-occurrence matrix, gradient map, Haar wavelet transform, and the autoregressive model for each of 9 color channels were computed (a total of 1629 image textures).

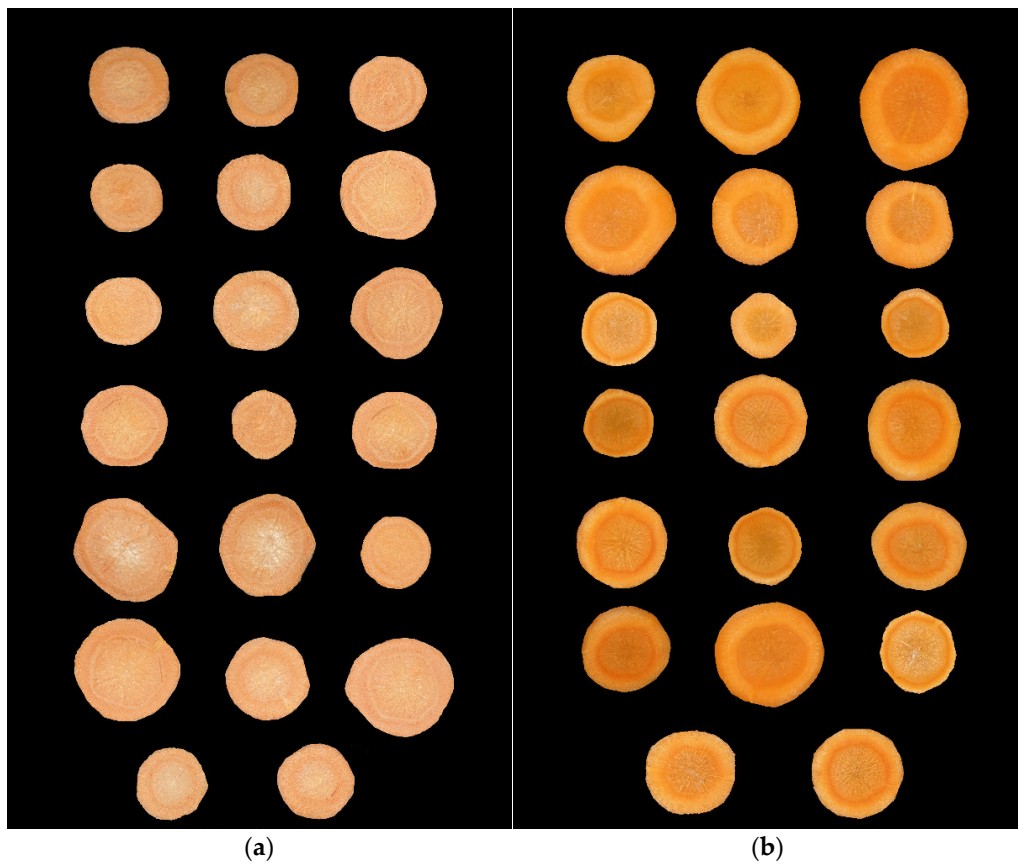

| (**a**) | (**b**) |

**Figure 3.** Sample color images of freeze-dried (**a**) and fresh (**b**) carrot slices.

### 2.3. Statistical Analysis

The determined texture parameters were used to build the models to distinguish freeze-dried and fresh carrot slices. The texture selection and model development were performed using WEKA machine learning software (Machine Learning Group, University of Waikato, Hamilton, New Zealand) [37–39]. Three methods of attribute selection, namely, the Genetic Search and CFS (Correlation-based Feature Selection) subset evaluator, the Ranker and Info Gain attribute evaluator, and the Best First and CFS subset evaluator, were used. In the case of each method, twenty texture parameters with the highest power for distinguishing the samples were selected from a set of textures from images in color channels *R*, *G*, *B*, *L*, *a*, *b*, *X*, *Y*, and *Z*. Various machine learning algorithms, from the groups of Trees, Rules, Meta, and Lazy, were applied to select the two most appropriate algorithms for each group, considering the highest accuracy. A test mode of 10-fold cross-validation was used to classify samples into freeze-dried and fresh carrot slices. That mode was characterized by a random division of the dataset into ten parts, treating each part in turn as the test set, and the remaining nine parts as the training sets. Thus, the learning procedure was carried out ten times on different training sets and the overall error was the average of ten error estimates. Ten-folds are considered appropriate to determine the best error estimate [40]. The confusion matrices with the number of correctly and incorrectly classified cases were determined. As other classification performance metrics,

average accuracies, Precision, F-Measure, MCC (Matthews Correlation Coefficient), ROC (Receiver Operating Characteristic) Area, and PRC (Precision-Recall) Area were computed. Furthermore, for selected models, the ROC (Receiver Operating Characteristic) curves and PRC (Precision–Recall) curves for the freeze-dried carrot and fresh carrot were determined. The presented results of the classification are for the 10-fold cross-validation mode.

## 3. Results

The results of the classification of freeze-dried and fresh carrot slices are presented separately for the models built based on image textures selected using the three methods, Genetic Search, Ranker, and Best First. Among the tested machine learning algorithms belonging to different groups, J48 and Random Forest from Trees, JRip and PART from Rules, Filtered Classifier and Multi-class Classifier from Meta, IBk and LWL from Lazy, and Logistic and SMO from Functions proved to be the most successful. For models developed using each of the selected algorithms, the confusion matrix, average accuracy, and the values of Precision, F-Measure, MCC, ROC Area, and PRC Area, are determined. Furthermore, for selected models, the ROC curves and PRC curves for the freeze-dried carrot and fresh carrot are shown.

*3.1. The Classification of Freeze-Dried and Fresh Carrot Slices, Based on Image Textures Selected using the Genetic Search*

The models for the classification of freeze-dried and fresh carrot samples were built based on twenty texture parameters selected using Genetic Search and the CFS (Correlation-based Feature Selection) subset evaluator. In the case of each color channel, image textures with the highest discriminative power were chosen, such as 2 textures from channel R, 3 textures from channel G, 2 textures from channel B, 4 textures from channel L, 2 textures from channel a, 2 textures from channel b, 2 textures from channel X, 1 texture from channel Y, and 2 textures from channel Z.

The developed models allowed for the classification of freeze-dried and fresh carrot samples, with an average accuracy reaching 100%, in the case of the Random Forest, Filtered Classifier, Multi-class Classifier, LWL, Logistic, and SMO machine learning algorithms (Table 1). All forty hundred cases were correctly classified. All two hundred cases belonging to the actual class of freeze-dried carrot slices were correctly included in the predicted class of freeze-dried carrot slices, and all two hundred cases from the actual class of fresh carrot slices were correctly classified as fresh carrot slices. The completely correct classification was confirmed by the Precision, F-Measure, MCC, ROC Area, and PRC Area equal to 1.000 for both classes (freeze-dried and fresh carrot slices). In the case of J48 and PART, average accuracy was equal to 99.75%, and only one case belonging to the freeze-dried samples was incorrectly classified as fresh ones. Freeze-dried and fresh carrot slices were distinguished, with an average accuracy of 99.5%, in the case of models built using JRip and IBk. However, the confusion matrices were different for these algorithms. The model developed using JRip produced 199 correctly classified cases for both classes, and only one incorrectly classified case for each class. The confusion matrix obtained by IBk was characterized by the completely correct classification of freeze-dried samples. In the case of fresh samples, 198 cases were correctly classified, and 2 cases were incorrectly included in the second class.

For selected models, the ROC (Receiver Operating Characteristic) curves and PRC (Precision–Recall) curves for the freeze-dried and fresh carrot slices were determined (Figure 4). In the case of models producing 100% correctness, the ROC and PRC curves for Random Forest were chosen to be presented. For both freeze-dried carrot slices and fresh carrot slices, curves present the ROC Area and PRC Area equal to 1.000 (Figure 4). For the models characterized by an average accuracy of 99.5%, the ROC and PRC curves for the IBk algorithm are presented (Figure 5). The curves created for freeze-dried samples show the ROC Area of 0.995 (Figure 5a), and PRC Area of 0.990 (Figure 5c). In the case

of fresh carrot slices, graphs for the ROC Area of 0.995 (Figure 5b) and PRC Area of 0.995 (Figure 5d) are presented.

**Table 1.** The performance metrics of the classification of freeze-dried and fresh carrot slices, based on models including image textures selected using the Genetic Search.

| Algorithm | Predicted Class | | Actual Class | Average Accuracy (%) | Precision | F-Measure | MCC | ROC Area | PRC Area |
|---|---|---|---|---|---|---|---|---|---|
| | Freeze-Dried | Fresh | | | | | | | |
| J48 | 199 | 1 | Freeze-dried | 99.75 | 1.000 | 0.997 | 0.995 | 0.998 | 0.998 |
| (Trees) | 0 | 200 | Fresh | | 0.995 | 0.998 | 0.995 | 0.998 | 0.995 |
| Random Forest | 200 | 0 | Freeze-dried | 100 | 1.000 | 1.000 | 1.000 | 1.000 | 1.000 |
| (Trees) | 0 | 200 | Fresh | | 1.000 | 1.000 | 1.000 | 1.000 | 1.000 |
| JRip | 199 | 1 | Freeze-dried | 99.5 | 0.995 | 0.995 | 0.990 | 0.995 | 0.993 |
| (Rules) | 1 | 199 | Fresh | | 0.995 | 0.995 | 0.990 | 0.995 | 0.993 |
| PART | 199 | 1 | Freeze-dried | 99.75 | 1.000 | 0.997 | 0.995 | 0.998 | 0.998 |
| (Rules) | 0 | 200 | Fresh | | 0.995 | 0.998 | 0.995 | 0.998 | 0.995 |
| Filtered Classifier | 200 | 0 | Freeze-dried | 100 | 1.000 | 1.000 | 1.000 | 1.000 | 1.000 |
| (Meta) | 0 | 200 | Fresh | | 1.000 | 1.000 | 1.000 | 1.000 | 1.000 |
| Multi-class | 200 | 0 | Freeze-dried | 100 | 1.000 | 1.000 | 1.000 | 1.000 | 1.000 |
| Classifier (Meta) | 0 | 200 | Fresh | | 1.000 | 1.000 | 1.000 | 1.000 | 1.000 |
| IBk | 200 | 0 | Freeze-dried | 99.5 | 0.990 | 0.995 | 0.990 | 0.995 | 0.990 |
| (Lazy) | 2 | 198 | Fresh | | 1.000 | 0.995 | 0.990 | 0.995 | 0.995 |
| LWL | 200 | 0 | Freeze-dried | 100 | 1.000 | 1.000 | 1.000 | 1.000 | 1.000 |
| (Lazy) | 0 | 200 | Fresh | | 1.000 | 1.000 | 1.000 | 1.000 | 1.000 |
| Logistic | 200 | 0 | Freeze-dried | 100 | 1.000 | 1.000 | 1.000 | 1.000 | 1.000 |
| (Functions) | 0 | 200 | Fresh | | 1.000 | 1.000 | 1.000 | 1.000 | 1.000 |
| SMO | 200 | 0 | Freeze-dried | 100 | 1.000 | 1.000 | 1.000 | 1.000 | 1.000 |
| (Functions) | 0 | 200 | Fresh | | 1.000 | 1.000 | 1.000 | 1.000 | 1.000 |

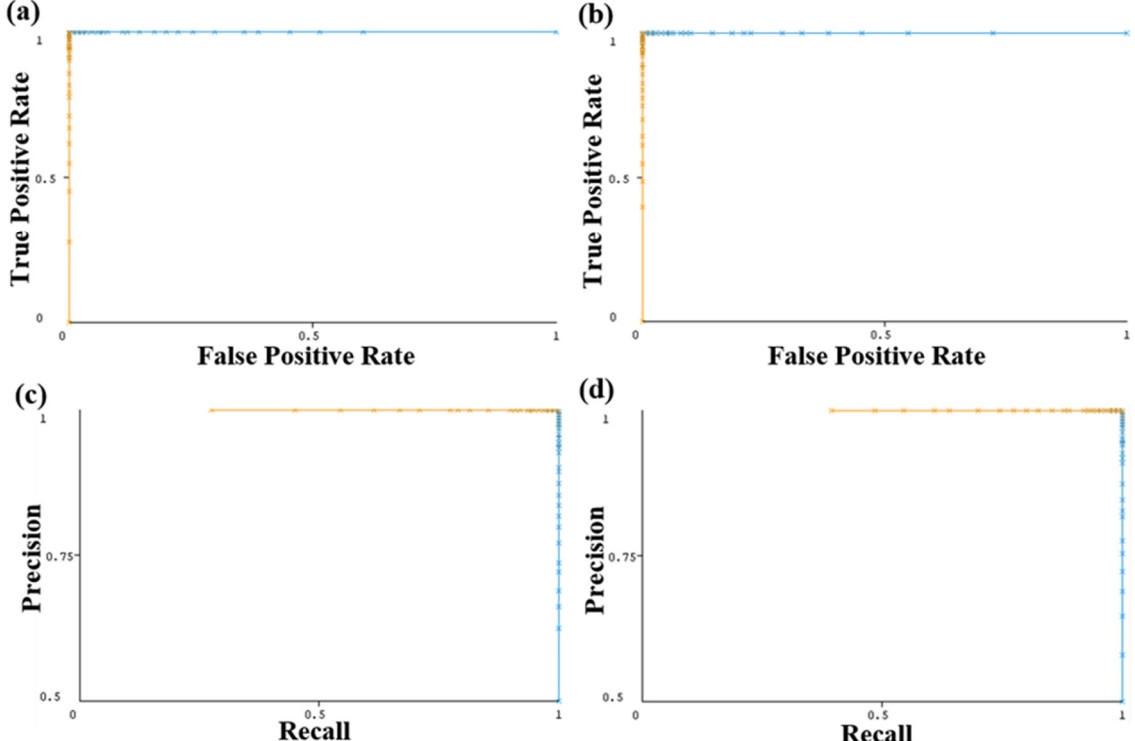

**Figure 4.** The ROC (Receiver Operating Characteristic) curves for the freeze-dried (**a**) and fresh carrot slices (**b**), and PRC (Precision–Recall) curves for the freeze-dried (**c**) and fresh carrot slices (**d**), for the model built using the Random Forest algorithm, based on image textures selected using the Genetic Search.

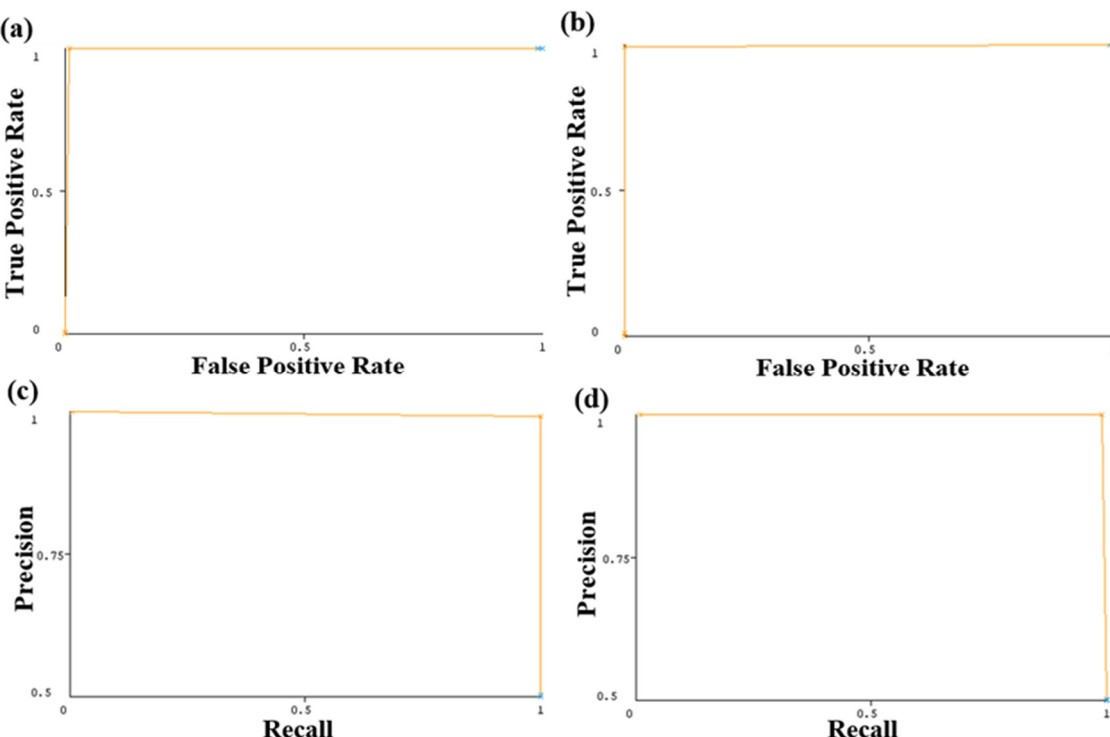

**Figure 5.** The ROC (Receiver Operating Characteristic) curves for the freeze-dried (**a**) and fresh carrot slices (**b**), and PRC (Precision–Recall) curves for the freeze-dried (**c**) and fresh carrot slices (**d**), for the model built using the IBk algorithm, based on image textures selected using the Genetic Search.

*3.2. The Distinguishing Freeze-Dried and Fresh Carrot Slices, Based on Image Textures Selected using the Ranker Search Method*

The most successful 20 image textures selected using the Ranker and Info Gain attribute evaluator were as follows: bHPerc10, bHPerc50, bHDomn10, bHMean, bHPerc99, bHDomn01, bHPerc90, BHPerc50, BHPerc10, BHMean, BHDomn01, ZHDomn10, BHDomn10, ZHDomn01, ZHPerc50, BHPerc90, ZHPerc10, ZHMean, ZSGPerc01, ZSGNonZeros. The selected textures allowed for distinguishing freeze-dried and fresh carrot slices, with an average accuracy of up to 100%, and the values of Precision, F-Measure, MCC, ROC Area, and PRC Area equal to 1.000 for models developed using Random Forest, Multi-class Classifier, IBk, Logistic, and SMO (Table 2). These results indicated the completely correct classification of carrot samples. The lowest average accuracy of 99.25% was observed for the model built using the JRip algorithm. It was confirmed by the results of the confusion matrix. In the case of freeze-dried carrot samples, 199 cases were correctly classified and 1 case was incorrectly included, in the class of fresh slices, whereas, 198 cases belonging to fresh carrot slices were correctly classified and 2 cases were incorrectly classified as freeze-dried ones. Among both classes, the freeze-dried carrot slices were characterized by lower values of Precision (0.990) and PRC Area (0.988). The value of F-Measure (0.992) was lower for fresh samples, whereas, the same values of MCC (0.985) and ROC Area (0.993) were found for freeze-dried and fresh carrot slices.

**Table 2.** The classification of freeze-dried and fresh carrot slices, using models based on image textures selected using the Ranker search method.

| Algorithm | Predicted Class | | Actual Class | Average Accuracy (%) | Precision | F-Measure | MCC | ROC Area | PRC Area |
|---|---|---|---|---|---|---|---|---|---|
| | Freeze-Dried | Fresh | | | | | | | |
| J48 | 199 | 1 | Freeze-dried | 99.75 | 1.000 | 0.997 | 0.995 | 0.998 | 0.998 |
| (Trees) | 0 | 200 | Fresh | | 0.995 | 0.998 | 0.995 | 0.998 | 0.995 |
| Random Forest | 200 | 0 | Freeze-dried | 100 | 1.000 | 1.000 | 1.000 | 1.000 | 1.000 |
| (Trees) | 0 | 200 | Fresh | | 1.000 | 1.000 | 1.000 | 1.000 | 1.000 |
| JRip | 199 | 1 | Freeze-dried | 99.25 | 0.990 | 0.993 | 0.985 | 0.993 | 0.988 |
| (Rules) | 2 | 198 | Fresh | | 0.995 | 0.992 | 0.985 | 0.993 | 0.990 |
| PART | 199 | 1 | Freeze-dried | 99.75 | 1.000 | 0.997 | 0.995 | 0.998 | 0.998 |
| (Rules) | 0 | 200 | Fresh | | 0.995 | 0.998 | 0.995 | 0.998 | 0.995 |
| Filtered Classifier | 200 | 0 | Freeze-dried | 99.75 | 0.995 | 0.998 | 0.995 | 0.998 | 0.995 |
| (Meta) | 1 | 199 | Fresh | | 1.000 | 0.997 | 0.995 | 0.998 | 0.998 |
| Multi-class | 200 | 0 | Freeze-dried | 100 | 1.000 | 1.000 | 1.000 | 1.000 | 1.000 |
| Classifier (Meta) | 0 | 200 | Fresh | | 1.000 | 1.000 | 1.000 | 1.000 | 1.000 |
| IBk | 200 | 0 | Freeze-dried | 100 | 1.000 | 1.000 | 1.000 | 1.000 | 1.000 |
| (Lazy) | 0 | 200 | Fresh | | 1.000 | 1.000 | 1.000 | 1.000 | 1.000 |
| LWL | 199 | 1 | Freeze-dried | 99.5 | 0.995 | 0.995 | 0.990 | 0.990 | 0.989 |
| (Lazy) | 1 | 199 | Fresh | | 0.995 | 0.995 | 0.990 | 0.990 | 0.984 |
| Logistic | 200 | 0 | Freeze-dried | 100 | 1.000 | 1.000 | 1.000 | 1.000 | 1.000 |
| (Functions) | 0 | 200 | Fresh | | 1.000 | 1.000 | 1.000 | 1.000 | 1.000 |
| SMO | 200 | 0 | Freeze-dried | 100 | 1.000 | 1.000 | 1.000 | 1.000 | 1.000 |
| (Functions) | 0 | 200 | Fresh | | 1.000 | 1.000 | 1.000 | 1.000 | 1.000 |

For models developed using image textures selected by the Ranker search method, the ROC and PRC curves for the Random Forest (Figure 6) and IBk (Figure 7) machine learning algorithms are presented with values of the ROC Area and PRC Area equal to 1.000 for both algorithms and both classes of freeze-dried and fresh carrot slices.

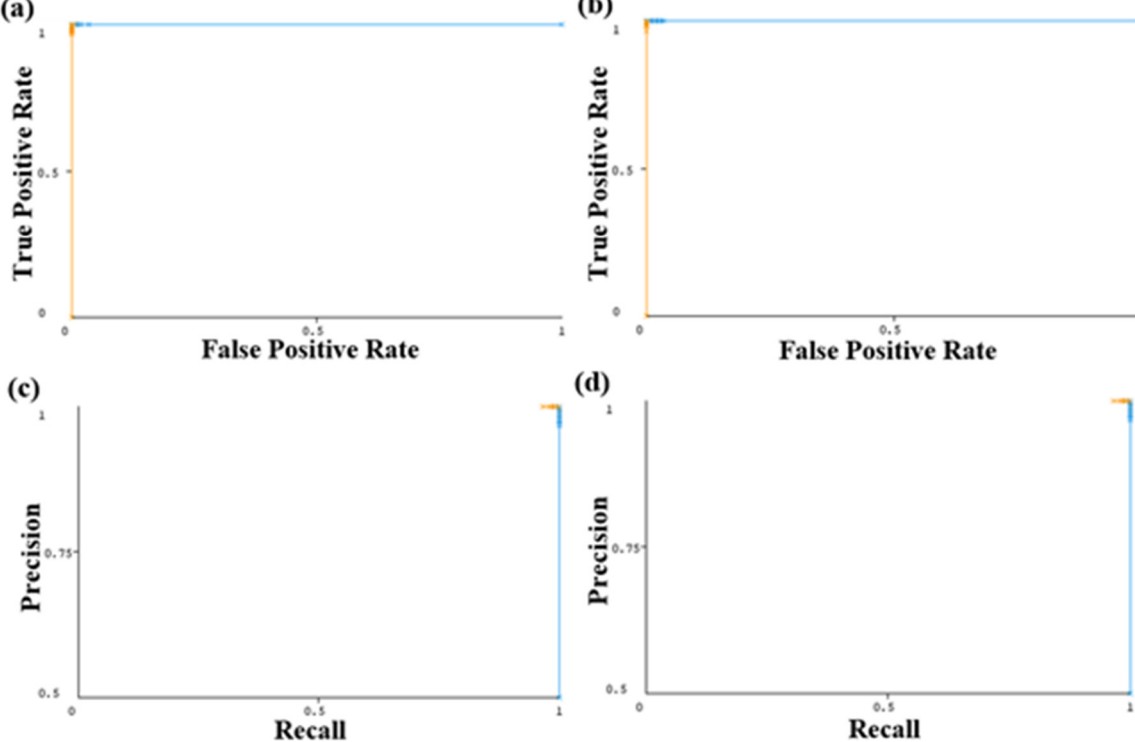

**Figure 6.** The ROC (Receiver Operating Characteristic) curves for the freeze-dried (**a**) and fresh carrot slices (**b**), and PRC (Precision–Recall) curves for the freeze-dried (**c**) and fresh carrot slices (**d**), for the model built using the Random Forest algorithm, based on image textures selected using the Ranker search method.

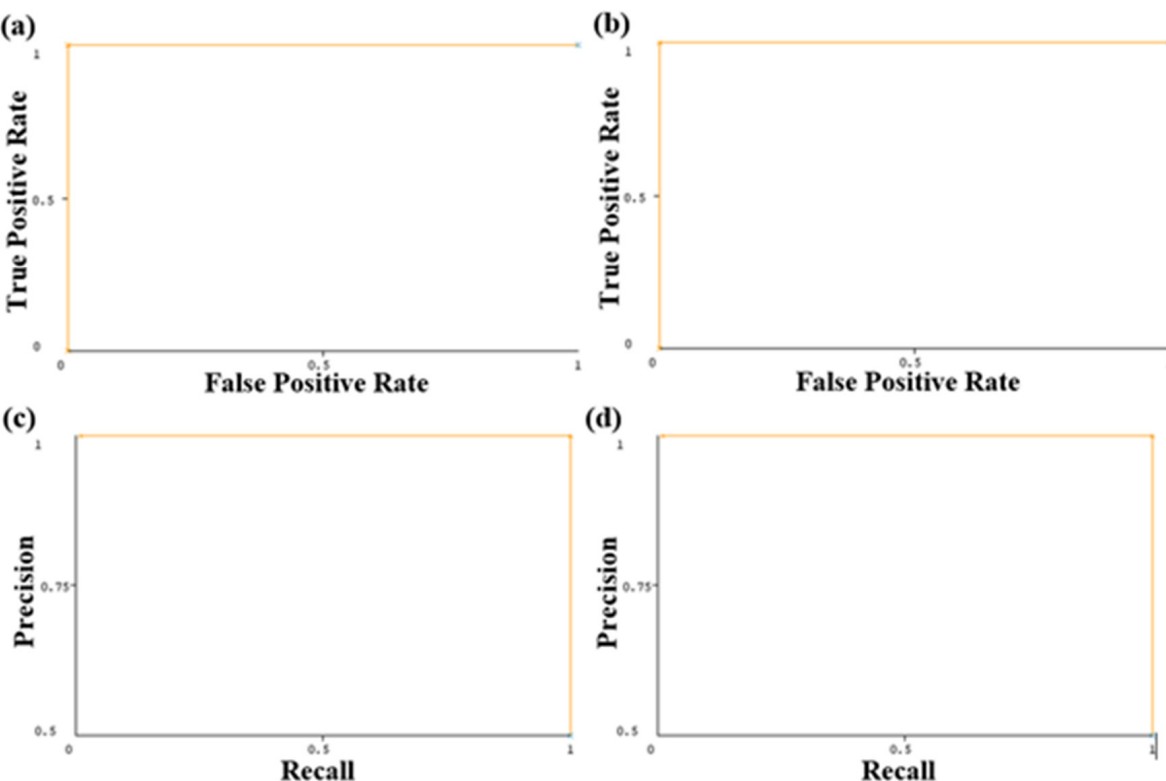

**Figure 7.** The ROC (Receiver Operating Characteristic) curves for the freeze-dried (**a**) and fresh carrot slices (**b**), and PRC (Precision–Recall) curves for the freeze-dried (**c**) and fresh carrot slices (**d**), for the model built using the IBk algorithm, based on image textures selected using the Ranker search method.

*3.3. The Classification of Freeze-Dried and Fresh Carrot Slices, Based on Image Textures Selected using the Best First Search Method*

The application of the Best First and CFS (Correlation-based Feature Selection) subset evaluator allowed for the selection of the following 20 image texture parameters with the highest discriminative power: RHDomn01, GHVariance, BHMean, BHPerc99, BS5SZ3SumOfSqs, bHMean, bHPerc01, bHPerc10, bHPerc50, bHPerc90, bHPerc99, bHDomn01, bHDomn10, bSGKurtosis, bS5SH1Contrast, bS4RZGLevNonU, YS5SZ5InvDfMom, ZHDomn01, ZS5SZ3InvDfMom, ZS5SZ5SumAverg. The models built based on selected image texture parameters produced an average accuracy of the classification of freeze-dried and fresh carrot slices ranged from 99.75% for J48, JRip, PART, and Filtered Classifier to 100% for Random Forest, Multi-class Classifier, IBk, LWL, Logistic, and SMO (Table 3). In the case of an average accuracy of 99.75%, 200 cases of one class, and 199 cases of the second class, were correctly classified, depending on the model. For models built using J48 and PART, fresh carrot slices were completely correctly distinguished from freeze-dried samples. In the case of freeze-dried slices, 199 cases were correctly classified, and one case was included in the class of fresh carrot slices. The JRip and Filtered Classifier algorithms produced a completely correct classification of freeze-dried carrot slices, whereas, 199 cases of fresh samples were correctly classified, and one case belonging to the actual class of fresh slices was incorrectly classified as freeze-dried carrot slices.

**Table 3.** The distinguishing freeze-dried and fresh carrot slices, using models based on image textures selected using the Best First search method.

| Algorithm | Predicted Class | | Actual Class | Average Accuracy (%) | Precision | F-Measure | MCC | ROC Area | PRC Area |
|---|---|---|---|---|---|---|---|---|---|
| | Freeze-Dried | Fresh | | | | | | | |
| J48 | 199 | 1 | Freeze-dried | 99.75 | 1.000 | 0.997 | 0.995 | 0.998 | 0.998 |
| (Trees) | 0 | 200 | Fresh | | 0.995 | 0.998 | 0.995 | 0.998 | 0.995 |
| Random Forest | 200 | 0 | Freeze-dried | 100 | 1.000 | 1.000 | 1.000 | 1.000 | 1.000 |
| (Trees) | 0 | 200 | Fresh | | 1.000 | 1.000 | 1.000 | 1.000 | 1.000 |
| JRip | 200 | 0 | Freeze-dried | 99.75 | 0.995 | 0.998 | 0.995 | 0.997 | 0.994 |
| (Rules) | 1 | 199 | Fresh | | 1.000 | 0.997 | 0.995 | 0.997 | 0.998 |
| PART | 199 | 1 | Freeze-dried | 99.75 | 1.000 | 0.997 | 0.995 | 0.998 | 0.998 |
| (Rules) | 0 | 200 | Fresh | | 0.995 | 0.998 | 0.995 | 0.998 | 0.995 |
| Filtered Classifier | 200 | 0 | Freeze-dried | 99.75 | 0.995 | 0.998 | 0.995 | 0.998 | 0.995 |
| (Meta) | 1 | 199 | Fresh | | 1.000 | 0.997 | 0.995 | 0.998 | 0.998 |
| Multi-class | 200 | 0 | Freeze-dried | 100 | 1.000 | 1.000 | 1.000 | 1.000 | 1.000 |
| Classifier (Meta) | 0 | 200 | Fresh | | 1.000 | 1.000 | 1.000 | 1.000 | 1.000 |
| IBk | 200 | 0 | Freeze-dried | 100 | 1.000 | 1.000 | 1.000 | 1.000 | 1.000 |
| (Lazy) | 0 | 200 | Fresh | | 1.000 | 1.000 | 1.000 | 1.000 | 1.000 |
| LWL | 200 | 0 | Freeze-dried | 100 | 1.000 | 1.000 | 1.000 | 1.000 | 1.000 |
| (Lazy) | 0 | 200 | Fresh | | 1.000 | 1.000 | 1.000 | 1.000 | 1.000 |
| Logistic | 200 | 0 | Freeze-dried | 100 | 1.000 | 1.000 | 1.000 | 1.000 | 1.000 |
| (Functions) | 0 | 200 | Fresh | | 1.000 | 1.000 | 1.000 | 1.000 | 1.000 |
| SMO | 200 | 0 | Freeze-dried | 100 | 1.000 | 1.000 | 1.000 | 1.000 | 1.000 |
| (Functions) | 0 | 200 | Fresh | | 1.000 | 1.000 | 1.000 | 1.000 | 1.000 |

## 4. Discussion

The obtained results confirmed the differentiation of image textures of freeze-dried and fresh carrot slices. Accuracy values for the Genetic Search feature selection method ranged from 99.5% to 100%. In this method, it has been determined that six different algorithms could classify with 100% accuracy. In addition, ROC Area values for this method were calculated to vary between 0.995 and 1.000. In the Ranker search feature selection method, accuracy values were found between 99.25% and 100%. This method has revealed that five different algorithms could classify with 100% accuracy. Moreover, ROC Area values for this method were found to vary between 0.990 and 1.000. Accuracy values for the Best First search feature selection method were obtained between 99.75% and 100%, and it was determined that six different algorithms were classified with 100% accuracy, and the remaining four algorithms could classify with 99.75%. ROC Area values were found to be between 0.994 and 1.000. Best First search has been chosen as the feature selection method that gives the most successful analysis results. Models built using machine learning algorithms classified both samples, with an average accuracy reaching 100% for selected algorithms. In machine learning, accuracy values may differ, depending on the balance of the data set, the size of the data set, and the algorithm parameters. Various machine learning algorithms from different groups have different characteristics and parameter settings [25,37–40]. The dependence of the correctness of carrot classification on the algorithm used was also demonstrated in the available literature. Xie, et al. [41] developed a carrot classification study based on machine learning methods. First, they extracted color (*R*, *G*, *B*, *H*, *S*, and *V*) and shape (circumference, diameter, aspect ratio, etc.) features from carrot images. They then fed these features into Artificial Neural Networks (ANN), SVM, and Extreme Learning Machine (ELM) algorithms. As a result of the study, the highest success was achieved by ELM, with 96.67%. The results of the study were found to be lower than the results of the present study. Xie, et al. [42] proposed a CNN-based CarrotNet model to classify carrots according to their defects. They created a dataset of 9290 samples for the experimental study. After preprocessing the images, the proposed deep network was fed with processed images. They classified deep features with four different machine learning algorithms (SVM, KNN, Random Forest (RF), Gaussian Naive Bayesian (GNB)) at the output of the network. At the end of the study, the highest accuracy obtained from the test data was 97.04%, in the step consisting of the combination of these four machine learning methods. This accuracy result was determined to be slightly lower

than the results of the similar algorithms (RF, SMO:SVM, IBK:KNN) applied in the current study. Jahanbakhshi and Kheiralipour [8] classified carrots based on their shape using machine learning methods. First, they extracted various shape features, such as width, perimeter, roundness, etc. from carrot images. Then, these features were classified by Linear Discriminant Analysis (LDA) and Quadratic Discriminant Analysis (QDA) methods. The accuracies obtained as a result of LDA and QDA were 92.59% and 96.30%, respectively. The results of the present study varied between 99.5% and 100% for different algorithms, and the results of the related study were higher. Zhu, et al. [43] generated image data from carrots of different physical shapes. The specific carrot features used for the experimental study were black spots, bent, normal, and fibrous roots. To classify this image data, they applied the AlexNet Convolutional Neural Network (CNN) model with the transfer learning method. As a result of the normal-abnormal (binary) and multiple classifications (four classes), they achieved 98.70% and 95.30% accuracy, respectively. In a different study, Zhu, et al. [44] extracted features from 12 different CNN models (AlexNet, ResNet18, VGG16, ResNet101, ShuffleNet, etc.) to classify carrot types according to their appearance, and classified these features with the Support Vector Machine (SVM) method, which is a machine learning algorithm. At the end of the study, ResNet101 + SVM model provided the highest accuracy, with 98.17%. The current study expanded the scope of using image processing and artificial intelligence to assess the quality of processed carrots. Further research may focus on distinguishing carrots processed with different techniques, using deep learning. The accuracy results of the present study were found to be higher than both the accuracy results (98.70% and 95.30%) of Zhu et al. [43] and the accuracy results (98.17%) of Zhu et al. [44].

## 5. Conclusions

The combination of image analysis and traditional machine learning allowed for completely correct distinguishing of freeze-dried and fresh carrot slices, based on image textures selected using different methods. The most successful algorithms were Random Forest, Multi-class Classifier, Logistic, and SMO, providing a classification accuracy reaching 100% for the sets of textures selected by each of the methods, such as Genetic Search, Ranker, and Best First. The obtained results confirmed the practical usefulness of the developed procedures to assess the effect of freeze-drying on the structure of the outer surface of carrot slices. The approach can be used in further studies involving other carrot-processing techniques. Evaluation of changes in the flesh of carrots can be performed at various stages of processing. In future studies, it is planned to determine the chemical parameters of carrot samples after a certain processing time, simultaneously with the calculation of texture parameters of the images. It may allow for the development of regression equations that would be useful in practice for estimating changes in the chemical properties of processed carrot samples over time using non-destructive procedures based on image analysis and machine learning. Therefore, the procedure combining image processing and machine learning can be of great practical application in carrot processing. Besides traditional machine learning, deep learning can also be used to classify fresh and processed samples.

**Author Contributions:** Conceptualization, E.R., K.S. and M.F.A.; methodology, E.R.; software, E.R.; validation, E.R. and K.S.; formal analysis, E.R.; investigation, E.R.; resources, E.R; data curation, E.R.; writing—original draft preparation, E.R., K.S., M.F.A. and N.Ç.; writing—review and editing, E.R., K.S. and M.F.A.; visualization, E.R., K.S; supervision, E.R. All authors have read and agreed to the published version of the manuscript.

**Funding:** The purchase of the freeze dryer was co-financed by the European Union through the European Regional Development Fund within the Innovative Economy Operational Programme, 2007–2013. Project No UDA–POIG.01.03.01-00-129/09-10, entitled: "Polish Trichoderma strains in plant protection and organic waste management", under Priority 1.3.1, subject area 'Bio'.

**Institutional Review Board Statement:** Not applicable.

**Informed Consent Statement:** Not applicable.

**Data Availability Statement:** The data presented in this study are available on request from the corresponding author.

**Conflicts of Interest:** The authors declare no conflict of interest.

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
