# Peer review of "Rapid Detection of Changes in Image Textures of Carrots Caused by Freeze-Drying using Image Processing Techniques and Machine Learning Algorithms"

_sustainability, doi:10.3390/su15087011_

Round 1

Reviewer 1 Report

I have no suggestions, the manuscript seems to me to be well structured and of good quality.

Author Response

Thank you very much for this comment.

Reviewer 2 Report

1. In the Introduction section, the drawbacks of each conventional technique should be described clearly.

2. The introduction needs to explain the main contributions of the work more clearly.

3. The Wide ranges of applications need to be addressed in Introductions.

4. The objective of the research should be clearly defined in the last paragraph of the introduction section.

5. Fig.2 should be redraw with all the machine learning algorithms used

6. What is the novelty of this research, I could not find any novelty in this

7. The motivation for the present research would be clearer, by providing a more direct link between the importance of choosing your own method.

8. In the results and discussion, the findings of the present research work should be compared with the recent work of the same field towards claiming the contribution made.

9. In Result section, author have used 10 algorithms and proved that RF is providing 100 % accuracy. Is practically possible? Justify your answer

10. Discuss about the computational analysis of the presented model.

Author Response

Comment 1. In the Introduction section, the drawbacks of each conventional technique should be described clearly.

Response 1.  Thanks for your valuable comments. The drawbacks of each conventional technique were added to the introduction section as follows:

 Since it is product exposed to open environments during open-sun drying, it can be easily polluted by dust, insects, rodents, and birds [12]. Although air convective drying is common and easy to apply, low product quality (loss of color and aroma) and high energy consumption limit the use of this technique [13]. In addition, in the convective method, shrinkage, and loss of nutrients and flavor may occur [14]. Microwave drying may cause over-browning of fruits and loss of taste due to exposure to excessive heat [15]. For this reason, freeze drying, which offers the best final product quality among drying methods, comes to the fore.

Comment 2. The introduction needs to explain the main contributions of the work more clearly.

Response 2.  Thanks for your valuable comments. It has been specified in the manuscript as follows:

 Therefore, the main contributions of this study are:

- detection of changes in carrots caused by freeze-drying in a non-destructive and objective manner,

- distinguishing fresh and dried carrot samples based on selected image textures,

- assessment of the usefulness of textures selected using various selection methods for the classification of fresh and dried carrot samples,

- comparison of the effectiveness of machine learning models built using different algorithms.

Comment 3. The Wide ranges of applications need to be addressed in Introductions.

Response 3.  Thanks for your valuable comments. They were addressed as follows:

A wide range of applications has been conducted on carrots using freeze-drying processes or methods [2, 14, 20-23].

2.            Lyu, Y.; Bi, J.; Chen, Q.; Li, X.; Wu, X.; Hou, H.; Zhang, X. Discoloration investigations of freeze‐dried carrot cylinders from physical structure and color‐related chemical compositions. Journal of the Science of Food and Agriculture 2021, 101, 5172-5181.

14.          Voda, A.; Homan, N.; Witek, M.; Duijster, A.; van Dalen, G.; van der Sman, R.; Nijsse, J.; van Vliet, L.; Van As, H.; van Duynhoven, J. The impact of freeze-drying on microstructure and rehydration properties of carrot. Food Research International 2012, 49, 687-693.

20.          Rawson, A.; Tiwari, B.; Tuohy, M.; O’donnell, C.; Brunton, N. Effect of ultrasound and blanching pretreatments on polyacetylene and carotenoid content of hot air and freeze dried carrot discs. Ultrasonics Sonochemistry 2011, 18, 1172-1179.

21.          Cui, Z.-W.; Li, C.-Y.; Song, C.-F.; Song, Y. Combined microwave-vacuum and freeze drying of carrot and apple chips. Drying Technology 2008, 26, 1517-1523.

22.          Fan, D.; Chitrakar, B.; Ju, R.; Zhang, M. Effect of ultrasonic pretreatment on the properties of freeze-dried carrot slices by traditional and infrared freeze-drying technologies. Drying Technology 2021, 39, 1176-1183.

23.          Rajkumar, G.; Shanmugam, S.; Galvâo, M.d.S.; Leite Neta, M.T.S.; Dutra Sandes, R.D.; Mujumdar, A.S.; Narain, N. Comparative evaluation of physical properties and aroma profile of carrot slices subjected to hot air and freeze drying. Drying Technology 2017, 35, 699-708.

Comment 4. The objective of the research should be clearly defined in the last paragraph of the introduction section.

Response 4.  Thanks for your valuable comments. The applied mode has been described in more detail as follows:

The objective of this study was to develop an innovative approach to classifying freeze-dried and fresh carrot slices using selected image texture parameters and machine learning algorithms. This is the first study that analyzes the change in freeze-drying processed carrot products with artificial intelligence-based methods in terms of textures selected from a large set of approximately 1600 features from images in color channels L, a, b, R, G, B, X, Y, and Z, which is the great novelty in evaluating differences between fresh and dried carrot. Different feature selection methods were used to choose textures with the highest power for distinguishing the carrot samples. Furthermore, various machine learning algorithms from Trees, Rules, Meta, Lazy, and Functions groups were applied.

Comment 5. Fig.2 should be redraw with all the machine learning algorithms used

Response 5. Thanks for your valuable comments. Figure 2 was redrawn with all machine learning algorithms used.

Figure 2. The block diagram of the classification of freeze-dried and fresh carrot slices using image processing and machine learning.

Comment 6. What is the novelty of this research, I could not find any novelty in this

Response 6. Thanks for your valuable comments. It has been corrected as follows:

This is the first study that analyzes the change in freeze-drying processed carrot products with artificial intelligence-based methods in terms of textures selected from a large set of approximately 1600 features from images in color channels L, a, b, R, G, B, X, Y, and Z, which is the great novelty in evaluating differences between fresh and dried carrot. Different feature selection methods were used to choose textures with the highest power for distinguishing the carrot samples. Furthermore, various machine learning algorithms from Trees, Rules, Meta, Lazy, and Functions groups were applied.

Comment 7. The motivation for the present research would be clearer, by providing a more direct link between the importance of choosing your own method.

Response 7.  Thanks for your valuable comments. The motivation for the present research was developing a procedure for distinguishing fresh and freeze-dried carrot samples in a non-destructive and objective manner. The approach will be used in further studies. It has been indicated in the manuscript as follows:

The obtained results confirmed the practical usefulness of the developed procedures to assess the effect of freeze-drying on the structure of the outer surface of carrot slices. The approach can be used in further studies involving other carrot processing techniques. Evaluation of changes in the flesh of carrots can be performed at various stages of processing. In future studies, it is planned to determine the chemical parameters of carrot samples after a certain processing time, simultaneously with the calculation of texture parameters of the images. It may allow the development of regression equations that would be useful in practice for estimating changes in the chemical properties of processed carrot samples over time using non-destructive procedures based on image analysis and machine learning. Therefore, the procedure combining image processing and machine learning can be of great practical application in carrot processing.

Comment 8. In the results and discussion, the findings of the present research work should be compared with the recent work of the same field towards claiming the contribution made.

Response 8.  Thanks for your valuable comments. The findings of the present research study were compared with recent studies in the same field (marked in red in the discussion section).

Comment 9. In Result section, author have used 10 algorithms and proved that RF is providing 100 % accuracy. Is practically possible? Justify your answer

Response 9.  Thanks for your valuable comments. This is possible because large amounts of data are emerging today that allow machines to be trained rather than programmed. Random Forest, one of the machine learning algorithms, has taken its place today as a technological approach that can analyze large amounts of data. Deeper discussions reveal the advantages of random forest resources from their strong learning capabilities, robustness, and feasibility of hypothesis space. Many studies (mentioned in the present study) have proven the superiority of random forest for regression modeling and daily applications (Ao et al., 2019).

Ao, Y., Li, H., Zhu, L., Ali, S., & Yang, Z. (2019). The linear random forest algorithm and its advantages in machine learning assisted logging regression modeling. Journal of Petroleum Science and Engineering, 174, 776-789.

Comment 10. Discuss about the computational analysis of the presented model.

Response 10.  Thanks for your valuable comments. It has been corrected as follows:

The obtained results confirmed the differentiation of image textures of freeze-dried and fresh carrot slices. Accuracy values for the Genetic Search feature selection method ranged from 99.5% to 100%. In this method, it has been determined that six different algorithms could classify with 100% accuracy. In addition, ROC Area values for this method were calculated to vary between 0.995 and 1.000. In the Ranker search feature selection method, accuracy values were found between 99.25% and 100%. This method has revealed that five different algorithms could classify with 100% accuracy. Moreover, ROC Area values for this method were found to vary between 0.990 and 1.000. Accuracy values for the Best First search feature selection method were obtained between 99.75% and 100%, and it was determined that six different algorithms were classified with 100% accuracy and the remaining four algorithms could classify with 99.75%. ROC Area values were found to be between 0.994 and 1.000. Best First search has been chosen as the feature selection method that gives the most successful analysis results. Models built using machine learning algorithms classified both samples with an average accuracy reaching 100% for selected algorithms. In machine learning, accuracy values may differ depending on the balance of the data set, the size of the data set and the algorithm parameters.

Reviewer 3 Report

The authors explored a method for quickly detecting texture changes in carrot images caused by freeze drying using image processing techniques and machine learning algorithms. This article is quite interesting and suitable for publication in Sustainability journal. The authors have already written quite well, and it is recommended that they refer to the MDPI format to adjust the line spacing of the references, and carefully proofread the article for typographical and grammatical errors. In my opinion, this manuscript can be directly published after careful proofreading of errors.

Author Response

The proofreading of errors was done.

Round 2

Reviewer 2 Report

Nil